# Source-Free Active Learning for Adapting Alzheimer's Diagnostic Deep Learning Models Across Neuroimaging Cohorts

Theofanis Ganitidis
School of Electrical and Computer Engineering
National Technical University of Athens
Athens, Greece
orcid.org/0009-0006-7794-9793
theogani@biosim.ntua.gr

Maria Eleftheria Vlontzou
School of Electrical and Computer Engineering
National Technical University of Athens
Athens, Greece
orcid.org/0009-0009-6557-3946

Maria Athanasiou
School of Electrical and Computer Engineering
National Technical University of Athens
Athens, Greece
orcid.org/0000-0003-1575-9100

Konstantina S. Nikita
School of Electrical and Computer Engineering
National Technical University of Athens
Athens, Greece
orcid.org/0000-0001-8255-4354

Christos Davatzikos
Centre for Biomedical Image Computing and Analytics
University of Pennsylvania
Philadelphia, PA, USA
orcid.org/0009-0009-6557-3946

*Abstract*—Alzheimer's Disease (AD) classification across multiple neuroimaging sites faces significant challenges due to domain shifts arising from variations in data acquisition protocols, imaging devices, and population demographics. While large-scale multi-site datasets offer unprecedented opportunities for developing robust diagnostic models, the heterogeneity between sites often leads to poor model generalization. This work proposes an uncertainty-informed active learning framework for Source-Free (SF) Domain Adaptation (DA) to classify cognitively normal individuals and AD patients across different neuroimaging studies. The proposed approach leverages Monte Carlo dropout to estimate prediction uncertainty and guide the selection of the most informative samples from the target domain for model adaptation, eliminating the need for source domain data during deployment. The framework was evaluated on a large-scale dataset comprising 3,177 participants from five neuroimaging studies (ADNI-1, ADNI-2/3, PENN, AIBL, and OASIS) with 145 regional brain volume measurements. The uncertainty-based active learning approach achieved the highest median AUC of 91.4% across all source-target combinations, outperforming baseline models (89.7%) and demonstrating superior performance compared to other SF and Source-Aware (SA) DA methods. Additionally, the distribution shifts between studies were quantified using maximum mean discrepancy to evaluate the effectiveness under variable inter-site shift. The results demonstrate that SF methods can achieve comparable or superior performance to SA approaches while addressing privacy constraints inherent in medical imaging applications.

*Keywords—domain adaptation, active learning, deep learning, uncertainty estimation, neuroimaging, Alzheimer's disease*

## I. INTRODUCTION

Alzheimer's Disease (AD) is the leading cause of dementia worldwide and is characterized by heterogeneous symptoms and progressive, irreversible cognitive decline [1], [2]. In recent years, collective efforts to unravel the complexity of AD have intensified, driving neuroimaging research to the era of large-scale multi-site analyses that integrate data across multiple and diverse cohorts, aiming to develop more advanced diagnostic and prognostic tools and to facilitate their adoption into clinical practice [3]. Leveraging large-scale neuroimaging data combined with the recent advances in machine learning (ML) and deep learning (DL) holds the potential to analyze complex, high-dimensional and heterogenous data towards the detection of subtle imaging signatures and new disease biomarkers, even at the pre-clinical stages of the disease [4].

However, the significant potential of large multi-site data comes with the challenge of poor generalization of ML models, which stems from the variations in data acquisition devices, imaging protocols and patient population distributions across sites [4], [5]. The heterogeneity of large multi-site training datasets, combined with the limited availability of annotated data at the deployment sites, hinders the clinical translation of AI-based decision support systems. To address the issue of limited reproducibility of ML models, a growing body of research has focused on developing domain adaptation (DA) techniques, designed to mitigate domain shifts across sites and tackle the challenge of model generalization [6].

Numerous DA approaches have been proposed, including few-shot DA, where models are trained using source domain data, along with a few labeled samples from the target domain. For instance, [7] uses a weighted empirical risk minimization (ERM) approach to optimally combine data from both domains for AD patient classification across different data sites. Another prominent category is unsupervised domain adaptation (UDA) that assumes annotated target domain data are unavailable, such as in [8], where a framework for inter-study DA is introduced that leverages auxiliary tasks based on readily available target domain covariates like age, gender or race to guide adaptation. In the context of DA, Active Learning (AL) represents a widely

adopted strategy in medical imaging for minimizing labeling costs while maximizing model performance. For example in [9], AL is employed to select informative samples from the target domain for training along with source domain data, using features that are consistent across both domains.

However, those DA methods rely on the unrealistic assumption that both the source and target domain data are accessible during the adaptation phase [10]. Due to privacy constraints, especially in the medical domain, there is a growing need for Source-Free (SF) DA techniques [11] that solely rely on target domain data and a model pre-trained on the source domain. For instance, in [12] a self-supervised approach is employed to generate pseudo labels for the target domain data and clustering is performed in the feature space to select the most informative samples from each pseudo-class, based on their proximity to the cluster centroids.

In this work, to enhance informative sample selection in the SF setting, the use of an uncertainty-informed AL framework is proposed for classifying healthy controls and AD patients across different data sites [13]. The present study leverages an AL approach that retrains the classification model with newly selected target domain samples. To guide the sample selection, model uncertainty is estimated using Monte Carlo (MC) dropout, enabling the ranking of target samples and the selection of the ones with highest uncertainty to effectively adapt the model to the target domain through model retraining. The contributions of the present work are threefold. First, it offers a comprehensive parameterization and evaluation of the proposed uncertainty-informed AL framework for AD diagnosis through extensive comparative experiments with existing DA methods. Second, a distribution shift quantification technique based on a two-sample test is applied to quantify the shift between data from different studies and investigate the impact of varying distribution shift levels on the performance gains achieved through DA. Finally, the study provides insights derived from a large-scale neuroimaging dataset characterized by significant variability in terms of data acquisition sites and population demographics.

## II. METHODS

Fig. 1 presents a schematic overview of the proposed uncertainty-informed AL framework. The process begins with pre-training a model on a labeled source domain, followed by fine-tuning using MC dropout to enable uncertainty estimation. An AL strategy is then employed to select the most informative samples from an unlabeled target domain dataset, guided by different uncertainty-based sampling methods. These selected samples are labeled or assigned pseudo-labels, and then used to further fine-tune the model, thereby enhancing its discriminative performance on the target domain. For completeness, the use of different SF and source domain-aware (SA) methods is also investigated and comparatively assessed with the proposed uncertainty-informed AL approach. Five datasets, presented in the following Section, are used for development and evaluation purposes. Each dataset is sequentially designated as the source domain, while the remaining datasets serve as target domains.

## A. Data

For this analysis, data were drawn from five studies of the iSTAGING dataset [14], including cognitively normal (CN) individuals and AD patients from the Alzheimer's Disease Neuroimaging Initiative (ADNI-1 and ADNI-2/3) [15], the Penn Memory Center and Aging Brain Cohort (PENN), the Australian Imaging, Biomarkers and Lifestyle (AIBL) [16], and the Open Access Series of Imaging Studies (OASIS) [17]. All brain scans underwent quality control (QC), which included visual inspection by a radiologist, and only baseline MRI sessions of participants with stable CN and AD diagnoses across all of their follow-ups were considered.

The dataset consisted of baseline cross-sectional volumetric measurements from T1-weighted MRI images across 145 ROIs for 3177 participants, of which 1941 were CN individuals and 1236 were AD patients. A summary of the demographic information across studies can be found in TABLE I. Preprocessing steps included correction for intensity inhomogeneities [18], removal of extra-cranial tissue via a multi-atlas skull stripping algorithm [19], and segmentation into ROIs using MUSE multi-atlas method [20].

## B. Model configuration

A DL model was developed and independently trained on each of the five neuroimaging studies, treating each study as a separate source domain. For the classification of CN and AD individuals in the source domain, the model architecture

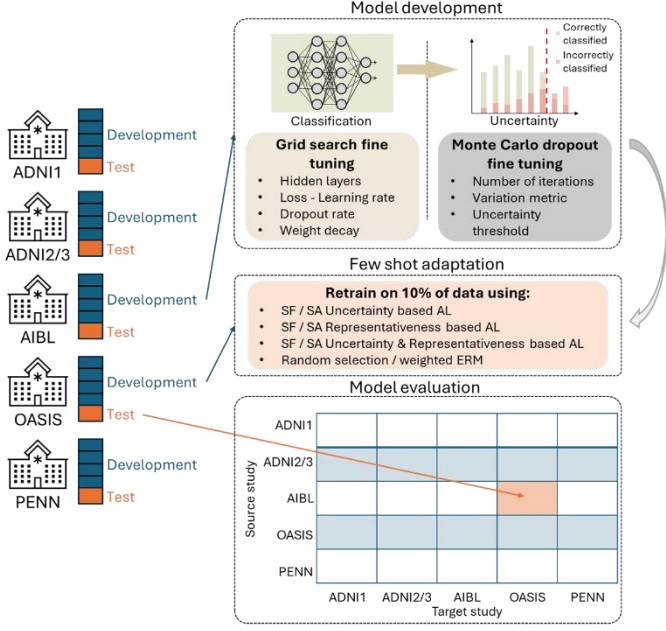

Fig. 1. Schematic overview of the proposed uncertainty-informed AL framework. Each study was split into 5 folds, 4 of which contributed to the model development and one was utilized for model evaluation. A baseline uncertainty-informed model was initially trained and validated on the development set and tested on the test set of the source study. MC dropout-based uncertainty estimation was deployed to guide the annotation and pseudo-labeling of selected traget domain samples, which were used for retraining the baseline models. In addition to the proposed AL framework, seven few shot adaptation methods, utilizing both source-free and source-aware approaches, were applied to the development set of the target study.

TABLE I.    DATASET SUMMARY

| | ADNI-1 | ADNI-2/3 | PENN | AIBL | OASIS | Total |
|---|---|---|---|---|---|---|
| **Subjects** | 11.1% | 22.6% | 12.3% | 22.6% | 31.4% | 100% |
| CN | 162 | 490 | 73 | 628 | 588 | 1941 |
| AD | 191 | 227 | 317 | 91 | 410 | 1236 |
| **Gender (%)** | | | | | | |
| Female | 48.4 | 55.4 | 63.3 | 59.9 | 57.1 | 57.2 |
| Male | 51.6 | 44.6 | 36.7 | 40.1 | 42.9 | 42.8 |
| **Age (%, years)** | | | | | | |
| <60 | 2.5 | 4.0 | 9.5 | 2.0 | 13.7 | 7.1 |
| 60-70 | 12.7 | 38.4 | 22.3 | 32.4 | 25.5 | 28.2 |
| 70-80 | 59.8 | 43.1 | 41.8 | 51.0 | 29.6 | 42.3 |
| >80 | 25.0 | 14.5 | 26.4 | 14.6 | 31.2 | 22.4 |
| **Race (%)** | | | | | | |
| White | 92.6 | 45.2 | 78.2 | 86.2 | 86.0 | 76.6 |
| Black | 5.4 | 3.2 | 15.6 | - | 12.0 | 7.0 |
| Other | 2.0 | 2.6 | 4.9 | - | 1.6 | 1.9 |
| Unknown | - | 49.0 | 1.3 | 13.8 | 0.4 | 14.5 |

consisted of multiple hidden layers, each comprising a tunable number of units, a rectified linear unit (ReLU) activation function, batch normalization, and dropout for regularization, while the output layer was a fully connected layer with sigmoid activation function and one node to support binary classification. To mitigate overfitting, early stopping based on the area under the ROC curve (AUC) was employed. The models were optimized using the Adam optimizer with a tunable learning rate. A 5-fold cross-validation scheme was used to configure the hyperparameters including the number of hidden layers, the number of units, the dropout rate, the learning rate, and the loss function used via random search.

## C. Adaptation to target domain

For the adaptation to the target domain, the models pre-trained on the source domain were retrained within the AL framework using the most informative samples from the target domain. Sample selection was guided by uncertainty ranking based on Monte Carlo dropout. High uncertainty samples were annotated, while low uncertainty ones were assigned pseudo-labels with variable weights in the loss function.

### 1) Uncertainty estimation with Monte Carlo dropout

Dropout layers are commonly used in neural networks to prevent overfitting by applying a binary mask to the model's input, randomly deactivating a subset of neurons during training. In contrast, MC dropout is applied during inference, to approximate Bayesian inference in deep Gaussian processes, allowing neural networks to estimate prediction uncertainty. This approach generates multiple sub-networks using different dropout masks and performs multiple independent forward passes for a given input, resulting in a set of outputs that form the distribution of predictions. The variability within this distribution of predictions reflects the model's uncertainty for those input samples [21], [22].

In the present framework, MC dropout was applied during inference on the target domain data, to identify informative samples for annotation that could be used to fine-tune the models pre-trained on the source domain. The hyperparameters of the MC dropout method were optimized on the source domain via cross-fold validation, and included the number of forward passes, the uncertainty threshold, and the metric for quantifying uncertainty. Several metrics were explored to assess prediction variability and quantify the uncertainty, including total variance, standard deviation, and predictive entropy - calculated as the sum of per-class variances. The uncertainty threshold served as the boundary between high and low uncertainty instances and was used to guide the selection of the low uncertainty samples for pseudo-labeling in the AL setting [13].

Assessing the uncertainty estimation method involved three steps. First, the model's predictions were calculated and compared to the ground truth labels, categorizing each source sample as correctly and incorrectly classified. Then, the AUC was calculated to assess the method's ability to discriminate between correctly and incorrectly classified samples. This metric was used to cross-validate the hyperparameters combinations. Finally, the optimal uncertainty threshold was calculated by maximizing the Youden's J statistic [23].

### 2) Uncertainty-informed AL

AL aims to minimize annotation effort by iteratively seeking the fewest possible and most informative samples from an unlabeled dataset for annotation, building on the notion that training a model on an optimally chosen subset can yield performance comparable to training on the entire dataset [24]. In this study, MC dropout was leveraged within AL for ranking target domain samples according to their estimated uncertainty [25]. Based on the uncertainty ranking, the most uncertain instances were selected for annotation. To ensure a fair comparison across different methods, the annotation ratio was fixed at 10% for all experiments, simulating clinician's feedback through the provision of ground truth labels. This controlled setting allowed for the isolation and systematic assessment of the effectiveness of the AL strategy without the influence of confounding factors such as dynamically varying annotation sizes. Among the remaining 90%, instances with low uncertainty, based on the experimentally defined uncertainty threshold, were assigned pseudo-labels. Both annotated and pseudo-labeled samples from the target domain were incorporated into the training procedure with differentiated contributions in the loss function, controlled by a weighting factor $a$ which was optimized through hyperparameter tuning to mitigate the impact of potentially misclassified samples [13]. Pseudo-labeled samples' weight was equal to $a$ while the annotated samples' weight was equal to $(1-a)$, with $a$ ranging from 0.1 to 0.5.

### 3) Comparison with source-free and source-aware DA approaches

A comparative assessment with widely used DA approaches was conducted to assess the effectiveness of the proposed uncertainty-informed AL framework. Within this experimental setup, the use of both SF and SA approaches was investigated:

i) *SF methods*: AL with random sampling of target domain samples was evaluated, followed by a representativeness-

based AL framework [12], and a combined approach that leverages both uncertainty and representativeness criteria. These strategies operate under a restrictive setting, assuming no access to the source domain data and relying only on a model pre-trained on the source domain.

ii) *SA methods*: A weighted ERM technique was applied [7], which retrains the model on the source domain data and on a few labeled samples from the target domain by assigning tunable weights to balance their contribution during training. Moreover, the SA alternatives of the aforementioned SF AL-based strategies were evaluated, which involved retraining on both the source domain data and selected samples of the target domain through applying uncertainty-based, representativeness-based, and combined criteria.

### D. Distribution shift quantification

A distribution shift quantification technique based on a two-sample test was applied to quantify the shift between data from different studies. An auxiliary model, incorporating the same hyperparameters as the baseline model, was trained and validated using the entire dataset to classify input instances to their corresponding clinical study (ADNI-1, ADNI-2/3, AIBL, OASIS, PENN). This approach helped to learn representations that preserved information specific to each clinical study, thus amplifying these study-specific features compared to the representations learnt from the main task. Maximum-mean discrepancy (MMD) was used as a two-sample test statistic on the learned representations to quantify the differences in the data distribution across instances from different studies.

## III. RESULTS

For the comprehensive evaluation of the proposed AL framework, each study among the ADNI-1, ADNI-2/3, PENN, AIBL, and OASIS was sequentially designated as the source domain, while the remaining studies served as target domain datasets, which resulted in the systematic assessment of all possible source-target domain combinations across the five studies. For each source-target study pair, a 5-fold cross validation scheme was applied, with discrimination performance evaluated based on the AUC score and its median across all source studies. Fig. 2 presents the AUC scores obtained for each target domain study across all source studies and DA methods. As it can be observed, although the baseline model generally achieved high AUC scores on all target domain studies for the various source-target domain combinations, it was consistently outperformed by the proposed uncertainty-based AL framework in all cases. Moreover, the proposed framework achieved the highest median AUC among SF methods for the AIBL, PENN, OASIS target studies, and performed comparably to random sampling-based AL and representativeness-based AL for the ADNI-1 and ADNI-2/3 studies, respectively. In addition, it outperformed all other methods, including the SA ones, for the PENN and OASIS target studies. Among the SA approaches, weighted ERM achieved the highest AUC score in the AIBL and ADNI-1 target studies, outperforming all other methods in these cases. For the PENN and ADNI-2/3 target studies, SA representativeness-

based AL yielded the highest AUC scores among SA methods, but only surpassed the proposed framework and other SF methods in the case of ADNI-2/3.

Fig. 3 summarizes the AUC scores obtained for all source-target domain study combinations across the various DA approaches. Uncertainty-based AL consistently delivered the highest

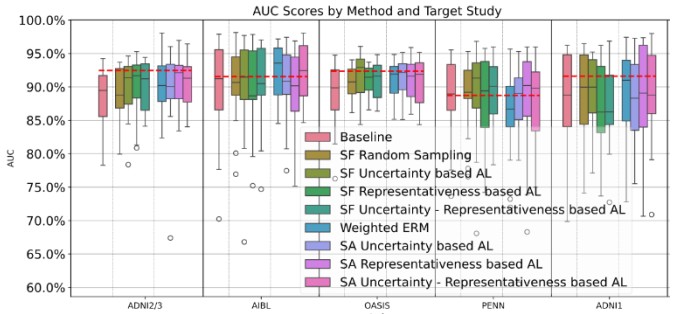

Fig. 2. AUC scores obtained for each target domain study. The first pink boxplot corresponds to the baseline case, where no adaptation was performed and the model was trained on a source study and evaluated directly on the target study. Results from the four SF AL-based approaches, followed by those from the four SA methods are shown with different colors. The dashed red line shows the performance achieved when the model was trained and evaluated on the same study, serving as an upper boundary.

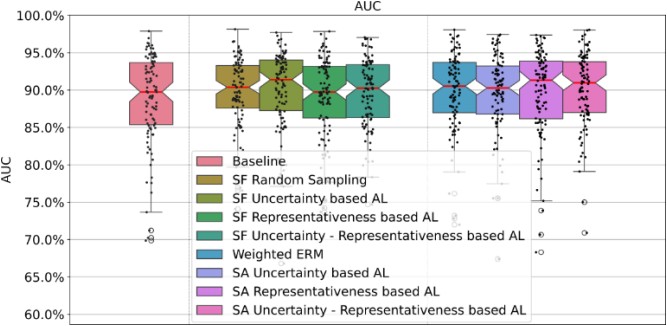

Fig. 3. Overall AUC scores across all source-target study combinations. The first pink boxplot corresponds to the baseline case, where no adaptation was performed and the model was trained on a source study and evaluated directly on the target study. Results from the four SF AL-based approaches, followed by those from the four SA methods are shown with different colors.

median AUC, reaching 91.4% across both SF and SA methods, closely followed by the SA version of representativeness-based AL that achieved a comparable median AUC of 91.3%, thus obtaining the highest score among SA approaches. All SA and SF methods outperformed the 89.7% median AUC of the baseline, except for SF representativeness-based AL that achieved the same median AUC with the baseline.

Despite the relatively high and comparable median AUC scores across all methods, ranging from 89.7-91.4%, paired statistical t-tests on the AUC scores between the baseline model and each DA approach for all source-target combinations revealed statistically significant performance differences. The p-values of the performed statistical t-tests are presented in TABLE II. For SF approaches, uncertainty-based AL and random sampling-based AL showed statistically significant differences with respect to the baseline model, with uncertainty-based AL

exhibiting a p-value of an order of magnitude lower. Among SA methods, only uncertainty-representativeness based AL and representativeness-based AL demonstrated statistically significant differences from the baseline model.

In terms of the distribution shift quantification, TABLE III. reports the MMD distances for each pair of clinical studies, calculated based on the representations extracted from the auxiliary model that was trained to predict the clinical study of a given input instance. The obtained mean MMD values for each study indicated variability in distributional divergence, suggesting that some studies were, in general, more shifted and others were more evenly distributed. Specifically, the PENN study demonstrated the highest degree of distribution shift, closely followed by ADNI-1 and AIBL, and finally ADNI-2/3 and OASIS.

To further investigate the impact of distributional divergence, the percentage change in the obtained AUC scores before and after adaptation was calculated. TABLE IV. shows the mean value of these changes for each source – target study combination across all DA methods, along with the mean change per source and target study, separately. Notable improvements in the obtained mean AUC scores across all target studies, reaching up to 7.2%, were observed when PENN served as the source study.

TABLE II.     STATISTICAL PAIRED T-TEST P-VALUES BETWEEN BASELINE AND DOMAIN ADAPTATION METHODS

| Domain Adaptation Method | Paired t-test p-value | |
|---|---|---|
| SA Uncertainty-Representativeness AL | 0.0018 | |
| SF Uncertainty-based AL | 0.0039 | |
| Random sampling-based AL | 0.0203 | |
| Weighted ERM | 0.0290 | |
| SA Representativeness-based AL | 0.0313 | 0.05 |
| SF Uncertainty-Representativeness AL | 0.0632 | |
| SF Representativeness-based AL | 0.2488 | |
| SA Uncertainty-based AL | 0.2742 | |

TABLE III.     MMD DISTANCES BETWEEN STUDIES.

| | ADNI-1 | ADNI-2/3 | AIBL | OASIS | PENN |
|---|---|---|---|---|---|
| ADNI-2/3 | 0.069 | | | | |
| AIBL | 0.101 | 0.008 | | | |
| OASIS | 0.049 | 0.005 | 0.015 | | |
| PENN | 0.015 | 0.074 | 0.107 | 0.045 | |
| **Mean** | 0.047 | 0.031 | 0.046 | 0.023 | 0.048 |

TABLE IV.     MEAN AUC DIFFERENCES BETWEEN BASELINE AND ADAPTED MODELS.

| Source \ Target | ADNI-1 | ADNI-2/3 | AIBL | OASIS | PENN | Mean |
|---|---|---|---|---|---|---|
| ADNI-1 | - | -0.7% | -0.5% | 0.8% | 2.9% | 0.6% |
| ADNI-2/3 | -1.2% | - | -1.9% | 0.4% | -0.3% | -0.7% |
| AIBL | -0.9% | 1.3% | - | 1.7% | 0.4% | 0.6% |
| OASIS | -0.4% | 0.7% | -1.6% | - | -2.7% | -1.0% |
| PENN | **3.4%** | **4.5%** | **7.2%** | **7.0%** | - | **5.5%** |
| Mean | 0.02% | 1.4% | 0.7% | 2.5% | 0.05% | - |

## IV. DISCUSSION

The comparative analysis revealed that DA methods consistently, though modestly, enhance the performance of models adapting to new target neuroimaging studies for the classification of CN and AD patients, using only MR imaging-derived features. The effectiveness and capacity of DA methods to improve model performance is a multifactorial outcome, influenced by the model's initial performance on the target domain, the specific characteristics of the target studies, differences between source and target domains, such as class imbalance or reversed class distributions, as well as the quality and representativeness of the data. The obtained performance gains were statistically significant (TABLE II. ) in the case of specific DA approaches and were achieved under cross-site generalization settings involving substantial distributional divergence (TABLE III. ), thus representing meaningful advancements in model reliability and robustness in the context of real-world clinical deployment, particularly when access to source data is restricted.

Interestingly, SF methods in many cases outperformed SA methods, demonstrating their potential to achieve comparable or even superior results, while eliminating privacy constraints emerging from the need for access to source domain data, as well as the cost of data annotation in the target domain. In particular, the proposed SF uncertainty-based AL framework achieved the highest performance based on the median AUC across all source-target combinations while yielding a more pronounced statistically significant performance improvement with respect to the baseline model than the best performing SA configurations. These results highlighted the proposed SF uncertainty-based AL method as an effective privacy-preserving approach, able to effectively address distribution shifts in AD classification under real-world deployment settings.

The distribution shift quantification analysis provided evidence regarding the relationship between domain similarity and adaptation effectiveness, offering valuable insights for future multi-site collaboration strategies. The comparison between the ranking of mean MMD distances and the ranking of mean performance changes revealed that studies exhibiting lower distributional shift tended to yield smaller or even negative performance gains when used as source domains, but showed greater improvements when used as target domains. This finding suggested that, regardless of the specific DA method or source (target) domain, the less (more) shifted the target (source) domain, the greater the benefit from the applied DA methods.

In terms of practical deployment, the proposed framework was computationally efficient, with full adaptation runs (including model tuning and active learning) completed in under one hour on an NVIDIA RTX 4060 GPU. The use of Monte Carlo dropout offered a practical trade-off between uncertainty estimation quality and computational cost, making the framework suitable for resource-constrained settings.

Potential limitations of the proposed work include the use of a fixed empirically chosen annotation threshold for AL. The fixed percentage approach may not be able to account for varying degrees of domain shift between different study pairs, which could lead to suboptimal resource allocation in some adaptation scenarios.

Several key areas for future research emerge from this work. Development of adaptive selection strategies that dynamically adjust the annotation ratio based on uncertainty distributions and domain shift characteristics could improve the efficiency of the active learning process. Future work could also investigate multi-task optimization approaches such as fairness-aware AL strategies that simultaneously maximize diagnostic accuracy and minimize demographic bias during the adaptation process. Finally, model trustworthiness could be enhanced through the retrospective assessment of pseudo-labeled samples and the incorporation of model interpretability techniques. The latter, would enable the identification of the most important brain regions, which beyond improving clinical transparency could also support the incorporation of real time clinician feedback to guide the adaptive sampling strategies in active learning.

## V. CONCLUSION

This study presented an uncertainty-informed active learning framework for SF domain adaptation in AD classification across multiple neuroimaging sites. The proposed approach successfully addressed the challenge of model generalization in multi-site neuroimaging studies while eliminating privacy constraints that limit data sharing between institutions. The comprehensive evaluation on 3,177 participants from five major neuroimaging studies demonstrated that the uncertainty-based AL method achieved superior performance across all source-target combinations, outperforming the baseline model. Moreover, the uncertainty-based AL method achieved matching or superior performance as compared to methods requiring access to source domain data, which highlighted the method's practical value for real-world deployment scenarios. The framework's ability to operate without access to source domain data makes it particularly suitable for clinical environments where data privacy and regulatory compliance are important concerns. By enabling robust model adaptation across diverse clinical sites without compromising data privacy, this method can serve as a practical, privacy-preserving AI solution for neuroimaging-based disease diagnosis.

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
