# OpenReview forum: "Source-Free Active Learning for Adapting Alzheimer’s Diagnostic Deep Learning Models Across Neuroimaging Cohorts"
_IEEE.org/EMBS/BHI/2025/Conference — BHI 2025_

### Official Review · Reviewer_psNG · 2025-07-04
**Source-Free Active Learning for Adapting Alzheimer’s Diagnostic Deep Learning Models Across Neuroimaging Cohorts**

**Confidence:** 5
**Clarity Of Writing:** great
**Clinical Significance:** great
**Methodological Novelty:** great
**Overall Rating:** 8

**Experiments And Results:**

excellent

**Questions For The Authors:**

Have you considered adaptive or dynamic thresholds for uncertainty-based annotation rather than a fixed top 10%?
	2.	How would this framework perform under severe class imbalance or noisy labels in the target domain?
	3.	Can you provide insights into the runtime and computational costs of using MC dropout during inference and AL iterations?
	4.	Is there any evidence or qualitative assessment of how the pseudo-labels compare to true annotations?
	5.	How do you envision incorporating clinician feedback or real-time labeling into this system in future work?

**Strengths:**

Privacy-aware modeling: Operates under a source-free setting, which is important in medical contexts with strict data sharing constraints.
	•	Robust experimental validation: Extensive evaluation on 3,177 subjects across five datasets using multiple DA methods (source-aware and source-free) with statistical testing.
	•	Uncertainty-informed AL: Employs MC dropout for uncertainty quantification, improving sample selection.
	•	Practical significance: The method is deployable in real-world clinical settings without source data, preserving patient privacy.
	•	Distribution shift quantification: Provides additional analysis using Maximum Mean Discrepancy (MMD), enhancing interpretability and reliability.

**Summary Of The Paper:**

This paper introduces an uncertainty-informed active learning (AL) framework for source-free domain adaptation (SFDA) to enhance Alzheimer’s Disease (AD) classification across multi-site neuroimaging datasets. It addresses the critical challenge of domain shift caused by heterogeneous acquisition protocols and population differences. The model estimates uncertainty using Monte Carlo (MC) dropout and selects the most informative unlabeled samples from the target domain for annotation or pseudo-labeling. The method is evaluated on five large neuroimaging datasets (ADNI-1, ADNI-2/3, PENN, AIBL, OASIS), achieving a median AUC of 91.4%, outperforming both source-free and source-aware baseline approaches.

**Weaknesses:**

Fixed annotation ratio: The use of a fixed top 10% uncertainty threshold for sample annotation might not adapt well across domains with varying shift magnitudes.
	•	Lack of interpretability: While performance improves, the model lacks a discussion on interpretability or clinical explainability, which is vital for healthcare adoption.
	•	Pseudo-labeling confidence tradeoff: There is limited discussion on potential risks of incorrect pseudo-labeling, especially when uncertainty estimation is imperfect.
	•	No comparison with non-DL techniques: The study excludes comparisons with classical ML or hybrid approaches which could offer a privacy-performance tradeoff.
	•	No human-in-the-loop validation: While the AL setup is discussed, the actual cost or feedback loop from clinicians isn’t modeled or simulated.

---

### Official Review · Reviewer_WmsH · 2025-07-14
**Source-Free Active Learning for Adapting Alzheimer’s Diagnostic Deep Learning Models Across Neuroimaging Cohorts**

**Confidence:** 2
**Clarity Of Writing:** great
**Clinical Significance:** good
**Methodological Novelty:** good
**Overall Rating:** 6
**Final Rating:** 7

**Experiments And Results:**

good

**Questions For The Authors:**

– Did you try adjusting the annotation ratio dynamically? A fixed rate might not work equally well everywhere.
– Any thoughts on runtime or how practical this is in settings with limited resources?
– Could this approach be extended to detect earlier stages, like MCI, or to handle more than two classes?

**Strengths:**

-The paper addresses a real clinical challenge data-sharing limits in neuroimaging by using a source-free setup, which feels both practical and timely.
-Smart use of uncertainty for sample selection: Monte Carlo dropout-based uncertainty estimation is well-justified and effective.
-The evaluation across five neuroimaging cohorts adds robustness to the findings.
-Includes statistical testing (paired t-tests), ablation comparisons with both SF and SA methods, and shift quantification.
-The method offers a privacy-preserving solution that is realistic for deployment in multi-site healthcare settings..

**Summary Of The Paper:**

This paper introduces an active learning (AL) framework that incorporates uncertainty into the process of source-free domain adaptation (SF-DA) for classifying Alzheimer’s disease (AD) based on neuroimaging data. The core challenge here is improving how well models generalize across different neuroimaging cohorts especially when the original source data can’t be accessed, which is quite common due to privacy restrictions.
To address this, the method relies on Monte Carlo (MC) dropout to estimate how uncertain the model is about its predictions. Based on this, it selects the most informative samples from the target domain to help guide the adaptation process. The experiments were done across five major datasets ADNI-1, ADNI-2/3, AIBL, OASIS, and PENN covering more than 3,000 participants and using 145 different regional brain volume features.
The results are quite promising: the framework achieved a median AUC of 91.4%, which is higher than several other approaches, including both source-aware and source-free domain adaptation methods. In addition, the paper looks into how distribution shifts between datasets can be quantified, using maximum mean discrepancy (MMD), and applies paired t-tests to confirm that the improvements are statistically significant.

**Weaknesses:**

- using a fixed 10% annotation rate for active learning seems a bit rigid some flexibility could help with different domain shifts.
- The performance improvement, while statistically valid, might not look that big (from 89.7% to 91.4%) to someone outside the domain adaptation field.
- There’s also not much insight into what brain regions drive the predictions, which could make it harder to apply clinically.
- And there’s no discussion about how heavy the method is computationally this might be an issue in real settings.

---

### Official Review · Reviewer_B6zq · 2025-07-17
**UAL for SFAD in AD classification**

**Confidence:** 3
**Clarity Of Writing:** good
**Clinical Significance:** good
**Methodological Novelty:** fair
**Overall Rating:** 5
**Final Rating:** 5

**Experiments And Results:**

good

**Questions For The Authors:**

- In II.D., it is mentioned that an auxiliary model is trained to classify clinical studies, which facilitated guiding the model to preserve and enhance information related to each clinical study. It is not clear to me how this has facilitated the guiding.
- In a similar theme as the previous question, it is not clear to me what the benefit of Table III is.
- Based on Figure 3, it is not so clear what the difference is between SA uncertainty-representativeness-based, representativeness-based, and SF uncertainty-based AL — which, up to a point, is presented in Table II.

**Strengths:**

- It presents a new UAL DA approach enabling more robust, generalized predictions.
- The proposed DA approach is SF, facilitating the deployment of pretrained models.

**Summary Of The Paper:**

This work proposes an uncertainty-informed active learning (UAL) framework for source-free domain adaptation (SFDA) to classify Alzheimer's disease (AD) and normal cases across five imaging studies. Using Monte Carlo Dropout (MCD), the method selects informative target-domain samples for model adaptation without requiring access to source data at deployment. The study focuses on evaluating the UAL framework with domain adaptation methods and a two-sample test-based technique to quantify distribution shift and analyze its impact on adaptation performance.

**Weaknesses:**

- Table I is in a very bad shape.
- Figures' quality can be improved.
- Figure 2 is very close to the text.
- It would be clearer if the difference were written mathematically.